# A First Approach towards Adsorption-Oriented Physics-Informed Neural Networks: Monoclonal Antibody Adsorption Performance on an Ion-Exchange Column as a Case Study

Vinicius V. Santana [1,2], Marlon S. Gama [3], Jose M. Loureiro [1,2], Alírio E. Rodrigues [1,2], Ana M. Ribeiro [1,2], Frederico W. Tavares [3,4], Amaro G. Barreto, Jr. [4] and Idelfonso B. R. Nogueira [1,2,*]

1   Laboratory of Separation and Reaction Engineering, Associate Laboratory LSRE/LCM Department of Chemical Engineering, Faculty of Engineering, University of Porto Rua Dr. Roberto Frias, 4200-465 Porto, Portugal; up201700649@edu.fe.up.pt (V.V.S.); loureiro@fe.up.pt (J.M.L.); arodrig@fe.up.pt (A.E.R.); apeixoto@fe.up.pt (A.M.R.)
2   ALiCE—Associate Laboratory in Chemical Engineering, Faculty of Engineering, University of Porto, Rua Dr. Roberto Frias, 4200-465 Porto, Portugal
3   Chemical Engineering Program—PEQ/COPPE, Universidade Federal do Rio de Janeiro (UFRJ), Cidade Universitária, Ilha do Fundão, Rio de Janeiro 21949-900, Brazil; marlongama@ufrj.br (M.S.G.); tavares@eq.ufrj.br (F.W.T.)
4   Chemical and Biochemical Engineering Processes (EPQB), School of Chemistry—EQ/UFRJ, Universidade Federal do Rio de Janeiro (UFRJ), Cidade Universitária, Ilha do Fundão, Rio de Janeiro 21949-900, Brazil; amaro@eq.ufrj.br
*   Correspondence: idelfonso@fe.up.pt

**Abstract:** Adsorption systems are characterized by challenging behavior to simulate any numerical method. A novel field of study emerged within the numerical method in the last two years: the physics-informed neural network (PINNs), the application of artificial intelligence to solve partial differential equations. This is a complete new standpoint for solving engineering first-principle models, which up to that date was not explored in the field of adsorption systems. Therefore, this work proposed the evaluation of PINN to address the numerical solutions of a fixed-bed column where a monoclonal antibody is purified. The PINNs solution is compared with a traditional numerical method. The results show the accuracy of the proposed PINNs when compared with the numerical method. This points towards the potential of this technique to address complex numerical problems found in chemical engineering.

**Keywords:** physics-informed neural networks; deep learning; phenomenological modeling; pressure swing adsorption

## 1. Introduction

Solving partial differential equation (PDE) systems is crucial in several fields. The PDEs are essential within engineering, ranging from civil engineering to chemical engineering. They are responsible for the mathematical representation of the observed reality through time and space. The more complex the relationship of the observed phenomena throughout these two dimensions, the more complicated the PDE systems are, and consequently, the more difficult it is to obtain their solution.

There are several numerical methods to address these problems. Following the nature of the PDEs, one can choose a myriad of options to obtain a numerical solution, ranging from the orthogonal colocation into finite elements to finite elements coped with Runge-Kutta. The numerical method is a consolidated field providing essential tools for engineers for several years. Recently, in 2019, this field started to be revolutionized by the advent of physics-informed neural networks (PINN) [1,2].

Using Artificial Neural Networks (ANN) to solve PDE systems is not new. It was first presented by [3]. The main idea is to leverage the universal approximation capacity of ANNs to solve PDEs. However, the method remained underused for many years due to the technology limitation. This strategy could be employed in practice only after the advent of deep learning and advanced computing techniques. [1] presented the first practical application of this concept.

The PINN can address complex numerical problems, usually a limitation of the classic numerical methods. For example, stochastic and high-order PDEs [1,4]. However, PINNs are based on the usage of the system conservation equations as constraints of a custom-designed loss function. Therefore, the design of the loss function plays a pivotal role in this technique. Following the peculiarities of the PDE system, supervised learning needs to be adequately designed. Thus, the PINNs methods should be carefully evaluated considering the application domain. As an emerging field of study, it still has limitations. PINN prediction is limited to the initial conditions assessed in the PDE/ODE solution.

Moreover, the PINN is initially obtained through a learning problem, which is computationally heavy. On the other hand, it is a very recent field of study with several potentials. For instance, once the system of PDEs is solved by PINNs, the resulting DNN can be used as a model. The trained PINN model can predict the system states in a fraction of milliseconds, which is an enormous advance in the field of rigorous simulation. Hence, it requires to be explored in the domain of application to evaluate its full potential. Therefore, this is one of the main motivations of this work, adapt this technique to the domain of chemical process, assessing its potentialities and limitations.

Chemical engineering problems are modeled with conservation and constitutive laws that yield complex PDE systems. The peculiarities of the phenomenological models that describe chemical processes usually take the classical numerical methods to their limits. Different length and time scales are present for several examples, challenging the PDE systems' solution. For instance, a system not limited by diffusion might present abrupt changes in its behavior, leading to problems solving the phenomenological model that describes this system. This is usual in processes that leverage adsorption to promote separation and/or reaction.

This work provides the first efforts in adsorption systems to apply PINN techniques to solve the numerical problem resulting from these systems' constitutive and conservative laws. We study the peculiarities of these systems from the point of view of physics-informed neural networks. Thus, we provide initial guidelines to address adsorption-based models using PINNs.

As a case study, an adsorption analysis of a therapeutic monoclonal antibody (mAb) is presented. It is known that chemical components and thermal stress lead to the formation of protein aggregates during mAbs production. A high-purity concentration has to be achieved if the aim is the use for a medicinal treatment [5]. Ion-chromatography columns are applied for the purification step to separate the mAbs from their impurities, and protein variants, due to the difference of electrostatic affinity between them and the ion-exchange stationary phase [6]. For simulating these systems, the electrostatic information must accurately describe the adsorption/desorption. Furthermore, simulating the adsorption behavior of mAbs in a column is a PDE problem, in which it is needed to solve a set of equations that contains the phenomenological representation of a fixed bed system. As an adsorption problem, it presents the abovementioned challenges to its numerical solution.

Thus, this work evaluates the application of PINNs to address the solution of a phenomenological model of a chromatographic column to purify a mAb protein using binding and elution strategy, as applied by [7]. It is important to note that adsorption and desorption usually yield different behaviors, which is a source of problems for the numerical solution of these systems. Therefore, it is a good case study to evaluate the proposed PINNs structure. This work paves the way to handle more complex problems in chemical engineering, namely, reactor modeling, gas-phase separation processes (pressure/temperature swing adsorption), and continuous chromatography (simulated moving bed).

## 2. Materials and Methods

### 2.1. Mathematical Model for Protein Adsorption in Column Chromatography

Column chromatographic process is usually modeled as a fixed bed by writing differential mass conservation laws in mobile and stationary phases. By solving these equations, one can predict the adsorbate concentration at any time and position. Considering that transport occurs predominantly in the axial direction ($x$), constant axial dispersion coefficient ($D_{ax}$), and the dimensionless mass balance in the mobile phase (bulk and boundaries) can be written as:

$$\frac{\partial C^*}{\partial t^*} + \frac{1 - \varepsilon_b}{\varepsilon_b} \frac{C_0}{q_0} \frac{\partial y}{\partial t^*} + \frac{\partial C^*}{\partial x^*} = \frac{1}{Pe} \frac{\partial^2 C^*}{\partial x^{*2}} \; ; x^* \in (0,1), t^* > 0 \tag{1}$$

$$\frac{\partial C^*}{\partial x^*} = Pe(C^* - C^*_{in}); \; x^* = 0, \; t^* \geq 0 \tag{2}$$

$$\frac{\partial C^*}{\partial x^*} = 0; x^* = 1, \; t^* \geq 0 \tag{3}$$

$$C^* = 0 \; (adsorption); x^* \in (0,1), \; t^* = 0 \tag{4}$$

$$y = \frac{K' C^*}{(1 + (K' - 1) C^*)}; \; K' - 1 = KC_0; \\ q_0 = \frac{q_{max} KC_0}{1 + KC_0} \tag{5}$$

$$C^* = \frac{C}{C_0}, \; x^* = \frac{x}{L}; \; t^* = t\frac{u}{L}; Pe = \frac{uL}{D_{ax}} \tag{6}$$

where $x^*$ is the dimensionless axial position, $t^*$ is the dimensionless time, $C^*$ is the dimensionless concentration, $y$ is the stationary phase dimensionless concentration in equilibrium with $C^*$ (mobile phase specie concentration), $Pe$ is the axial Peclet number, $\varepsilon_b$ is the bed porosity, $u$ is the interstitial velocity, $L$ is the column length, $C_0$ is a reference mobile phase concentration (inlet or feed), $q_0$ is a reference stationary phase concentration, $C^*_{in}$ is the dimensionless concentration at the column inlet, $C^*_0$ is the dimensionless concentration at initial condition, $q_{max}$ the maximum adsorption capacity of the stationary phase, $K\prime$ is the constant separation factor and $K$ is the dimensionless adsorbate-adsorbent interaction parameter given by:

$$K = \frac{A_s}{1 - \varepsilon_b} \int_0^\infty \left[ exp\left( -\frac{W(I, pH, h)}{k_b T} \right) - 1 \right] dh, \tag{7}$$

where $W$ is the binding free energy, $A_s$ is the accessible surface area per unit of packed bed volume ($4.2 \times 10^7 \, \mathrm{m}^{-1}$) [8], $I$ is the ionic strength, $h$ is the distance between the mAb and the stationary phase surfaces, $k_b$ is the Boltzmann constant and $T$ is the temperature.

The adsorption equilibrium constant is a thermodynamic property, a function of $W$ (potential of mean force, i.e., PMF), related to the adsorption binding affinity. It is noteworthy that electrostatic interaction is the leading property designating biomolecule surface interactions.

The literature [9,10] reports how to calculate the PMF from the electrostatic potential obtained by solving a modified Poisson-Boltzmann (PB) equation in a different set of coordinates. The modification in the PB equation allows the PMF to be a function of the solution conditions, i.e., temperature, ionic strength, salt type, and pH.

It is beyond the purpose of this work to show how to solve the modified PB equation. Thus, we applied the exact calculation details reported in the literature [9,10]. In this way, the Langmuir isotherm (Equation (5)) is linked with a thermodynamic parameter (Equation (7)), becoming a function of the mAb and stationary phase's surface properties and the solution conditions.

### 2.2. Simulation Scenario

A simulation of an adsorption-based separation is considered here to assess the PINN performance. Hence, the adsorption of a mAb with known charge density taken from [11] over YMC BioPro SP (10 μm) where the solid phase is initially protein-free is evaluated. The liquid phase is at a constant pH of 5.9, ionic strength 100 mM, and a feed stream at a concentration of 15 g/L is fed at the column inlet. Figure 1 describes the antibody protein ion-exchange adsorption schematically. The figure presents the ions and the protein/adsorbent surfaces charged, depicting the system mechanisms.

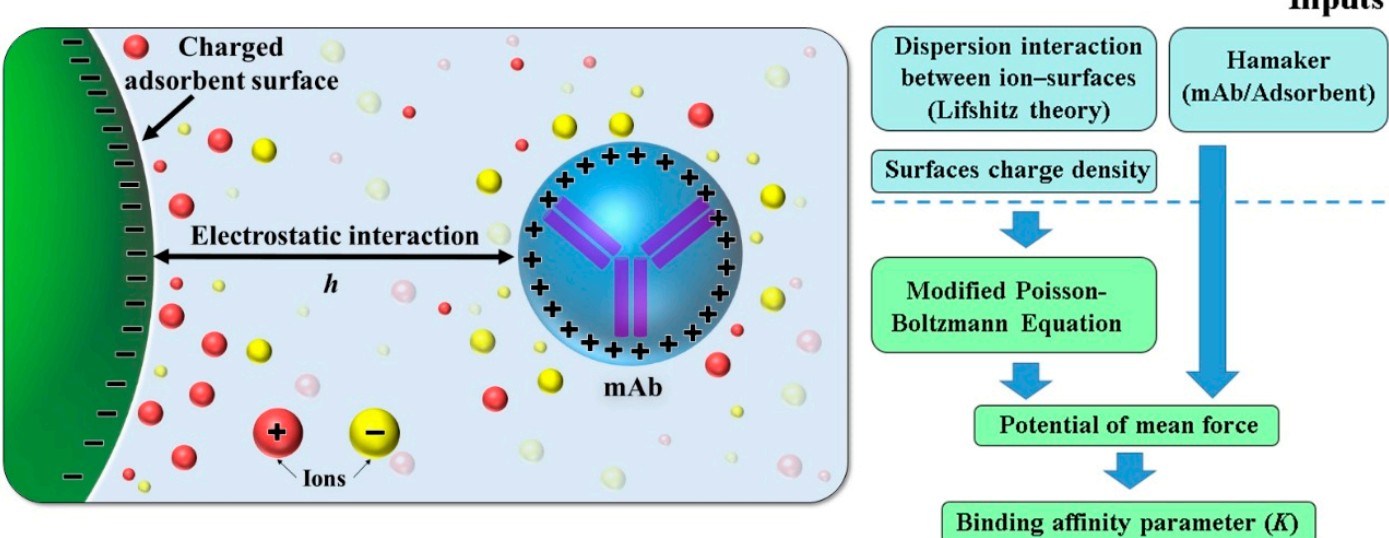

**Figure 1.** Schematical representation of the antibody adsorption by ion-exchange.

Under these conditions, the adsorption equilibrium constant ($K$, from Equation (5)) was estimated as 4.004 by solving the modified PB equation, as previously mentioned [9].

The chosen value for $K$ aims to represent an intermediary condition in protein adsorption—in a typical protein separation scenario (where high efficiency is required) $K$ should be much higher during adsorption and much lower during desorption. However, the scope of this work is comparing traditional numerical methods with a new approach (PINNs), and the chosen value for $K$ fits this purpose well. Moreover, the maximum adsorption capacity ($q_{max}$) is set as 125 g/L for the referred protein-adsorbent pair.

We used the following hydrodynamic parameters reported in the literature [8] as a reference for mAb performance in ion-exchange columns, namely: the bed porosity ($\varepsilon_b$), column length, column diameter, injection volume, and axial Peclet number ($Pe$) were set to be 0.53, 8.5 cm, 0.5 cm, 1.2 cm$^3$, and 500, respectively.

The solution of the fixed bed model via the Method of Lines (MoL) was implemented in a gPROMs Model Builder advanced process simulator with 60 finite elements and 3 collocation points per element. The resulting differential-algebraic system of equations was solved with the default DASOLV solver and automatic index reduction algorithm available in gPROMs. The physics-nformed neural network was built and trained in the DeepXDE [12] version 0.14.0.

### 2.3. Physics Informed Neural Networks

Given a generic space, a time-dependent nonlinear partial differential equation in the form:

$$u_t + \mathcal{N}(u) = 0; x \in \Omega; t \in [0, T], \tag{8}$$

and boundary conditions with nonlinear differential operator $B[\cdot]$:

$$\mathcal{B}(u, x, t) = 0 \text{ on } \partial\Omega \tag{9}$$

where $u(t, x)$ denotes the latent solution of the PDE, $u_t$ is the time derivate of $u$ and $\mathcal{N}[\cdot]$ is the nonlinear differential operator, one can approximate the unknown PDE solution $u(x, t)$ by a deep neural network $\hat{u}(x, t, \boldsymbol{\theta})$ with trainable parameters $\boldsymbol{\theta}$ and derive the called physics-informed neural networks [1,13] $f(\hat{u}, x, t)$ and $g(\hat{u}, x, t)$ as:

$$f(\hat{u}, x, t) = \hat{u}(x, t, \boldsymbol{\theta})_t + \mathcal{N}(\hat{u}(x, t, \boldsymbol{\theta})) \tag{10}$$

$$g(\hat{u}, x, t) = \mathcal{B}(\hat{u}, x, t) \tag{11}$$

Then, one can train the neural network $\hat{u}(x, t)$ to satisfy the $f(\hat{u}, x, t)$, $\mathcal{B}(\hat{u}, x, t)$ and initial condition constraints.

In practice, a set of points inside the space-time domain $(\mathcal{T}_f)$, another set in the boundaries $(\mathcal{T}_b)$ and at initial condition $(\mathcal{T}_i)$, are sampled and used to calculate the overall loss function $\mathcal{L}(\boldsymbol{\theta}, \mathcal{T})$ as:

$$\mathcal{L}(\boldsymbol{\theta}, \mathcal{T}) = w_f \mathcal{L}_f\left(\boldsymbol{\theta}; \mathcal{T}_f\right) + w_b \mathcal{L}_b(\boldsymbol{\theta}, \mathcal{T}_b) + w_i \mathcal{L}_i(\boldsymbol{\theta}, \mathcal{T}_i), \tag{12}$$

where,

$$\mathcal{L}_f\left(\boldsymbol{\theta}; \mathcal{T}_f\right) = \frac{1}{\left|\mathcal{T}_f\right|} \sum_{x,t \in \mathcal{T}_f} |f(\hat{u}, x, t)|^2 \tag{13}$$

$$\mathcal{L}_b(\boldsymbol{\theta}, \mathcal{T}_b) = \frac{1}{|\mathcal{T}_b|} \sum_{x,t \in \mathcal{T}_b} |g(\hat{u}, x, t)|^2 \tag{14}$$

$$\mathcal{L}_i(\boldsymbol{\theta}, \mathcal{T}_i) = \frac{1}{|\mathcal{T}_i|} \sum_{x,t \in \mathcal{T}_i} |\hat{u} - u(x, 0)|^2 \tag{15}$$

Figure 2 depicts the physics-informed neural network (PINN) building process to find the solution of differential equations.

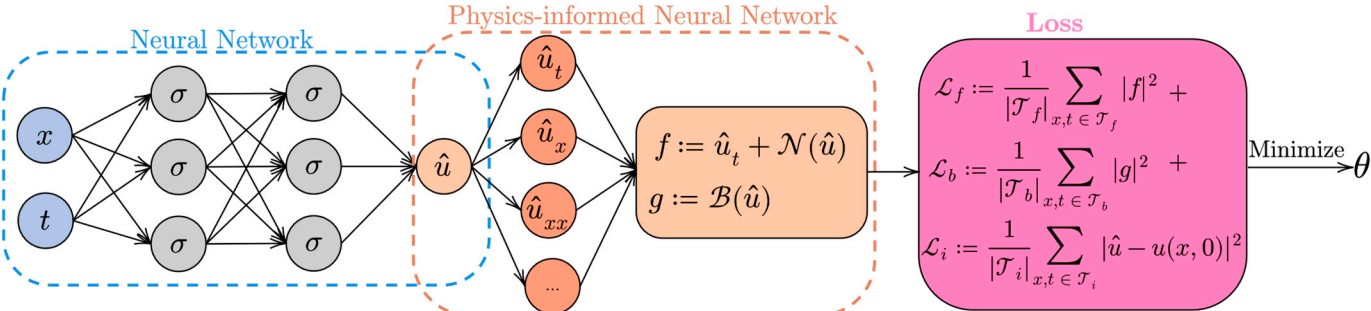

**Figure 2.** Physics-informed neural network building process.

Hence, considering the problem of protein adsorption in a fixed bed column, we propose the loss functions:

$$\mathcal{L}_f = \frac{1}{\left|\mathcal{T}_f\right|} \sum_{x,t \in \mathcal{T}_f} \left| \frac{\partial \hat{u}}{\partial t^*} + \frac{1-\varepsilon}{C_0 \varepsilon} \frac{\partial}{\partial t^*}\left(\frac{K\hat{u}C_0}{1 + \frac{K\hat{u}C_0}{q_{max}}}\right) + \frac{\partial \hat{u}}{\partial x^*} - \frac{1}{Pe} \frac{\partial^2 \hat{u}}{\partial x^{*2}} \right|^2 \tag{16}$$

$$\mathcal{L}_{bcl} = \frac{1}{|\mathcal{T}_b|} \sum_{x,t \in \mathcal{T}_b} \left| \frac{\partial \hat{u}(0, t)}{\partial x^*} - Pe(\hat{u}(0, t) - C_{in}^*) \right|^2 \tag{17}$$

$$\mathcal{L}_{bcr} = \frac{1}{|\mathcal{T}_b|} \sum_{x,t \in \mathcal{T}_b} \left| \frac{\partial \hat{u}(1, t)}{\partial x^*} \right|^2 \tag{18}$$

$$\mathcal{L}_i = \frac{1}{|\mathcal{T}_i|} \sum_{x,t \,\in \mathcal{T}_i} |\hat{u}(x,0) - C_0^*|^2 \tag{19}$$

By minimizing the sum of Equations (17)–(19) with an appropriate optimizer or a combination of optimizers, one can find the neural network $\hat{u}(x,t,\boldsymbol{\theta})$ parameters $\boldsymbol{\theta}$ that minimize the loss, leading to the solution of the PDE.

## 3. Results

This section provides the numerical experiment to assess the PINNs performance. The proposed solution is compared with the current state-of-the-art numerical method (MoL). Usually, these models are solved using MoL with axial position discretized with orthogonal collocation on finite elements.

For the numerical adsorption experiment, the feedforward architecture was employed. The hyperparameters, i.e., layers and neurons, were searched via random grid search. The best architecture, i.e., with minimum training error, had six layers with 80 neurons per layer. Layer-wise, locally adaptive hyperbolic tangent activations proposed in [14] were used in each layer to improve convergence speed. The weights of the loss function were set equal to 1.0 for all terms except for the boundary condition term at axial position $z = 0$. For this term, the corresponding weight ($W_{bcl}$) was adaptively changed during training, starting at $1 \times 10^{-4}$ and then finishing at 1.0 due to the associated high loss at the beginning of the training compared to other terms. An additional term was included in the loss function, namely, the derivative of the PDE residuals with respect to both dimensionless time and position ($t^*$, $x^*$) weighted by $1 \times 10^{-4}$ as it was previously shown to improve training loss [14].

A total of 2500 collocation points were used for the PDE residual ($\mathcal{T}_f$), 800 for both boundary conditions ($\mathcal{T}_b$), and 500 for initial condition ($\mathcal{T}_i$). The points were sampled using the Latin hypercube sampling strategy [15]. Then, the loss function was minimized by running 25,000 iterations of Adam [16] algorithm at a $3.0 \times 10^{-4}$ learning rate with early stopping (5000 iterations patience) followed by L-BFGS-B [17] until convergence.

Table 1 summarizes the neural network hyperparameters for the numerical adsorption experiment.

**Table 1.** Parameters to define the neural network for the adsorption experiment.

| Parameter | Description | Value |
|:---:|:---:|:---:|
| $N_l$ | Number of layers | 6 |
| $N_n$ | Number of neurons per layer | 80 |
| $w_f$, $w_{bcl}$, $w_{bcr}$, $w_i$ | Weights of terms in the loss function | (1.0, 1.0, adaptive,1.0) |
| $\mathcal{T}_f$, $\mathcal{T}_b$, $\mathcal{T}_i$ | Collocation points of PDE, boundary, and initial condition | (2500, 800, 500) |
| $\alpha$ | Learning rate of Adam algorithm | $3.0 \times 10^{-4}$ |

Figure 3 displays the training history on a semi-log scale. It shows that the loss starts decreasing fast and monotonically and then stagnates at around 0.1. When L-BFGS-B algorithm is used, a pronounced drop in the loss is reached until it reaches the final loss (order of $10^{-4}$).

Figure 4 compares the MoL and PINN solutions from three different perspectives. The first (Figure 4a) shows the solutions and error fields as a color map in the dimensionless space-time ($x^*$, $t^*$) domain, i.e., for every tuple ($x^*$, $t^*$), the solution of the PDE can be read by matching the corresponding color at the color bar in the Figures' right-hand side. The rightmost figure shows the absolute difference between the solution field obtained by PINN and by MoL. It can be seen that the PINN could learn PDE solution satisfactorily with the largest errors (mean absolute error) found in the breakthrough region, i.e., the line where the inlet concentration reaches these points. Figure 4b shows the concentration profile at six different dimensionless times. It is possible to see that at higher times and closer to the upper boundary ($x^* = 1$) the error is more pronounced; however, the profiles are satisfactorily learned by the PINN model. Figure 4c shows the history of the concentration

at several axial positions. It shows that the concentration changes suddenly in a short period, characterizing a compressive concentration front propagation.

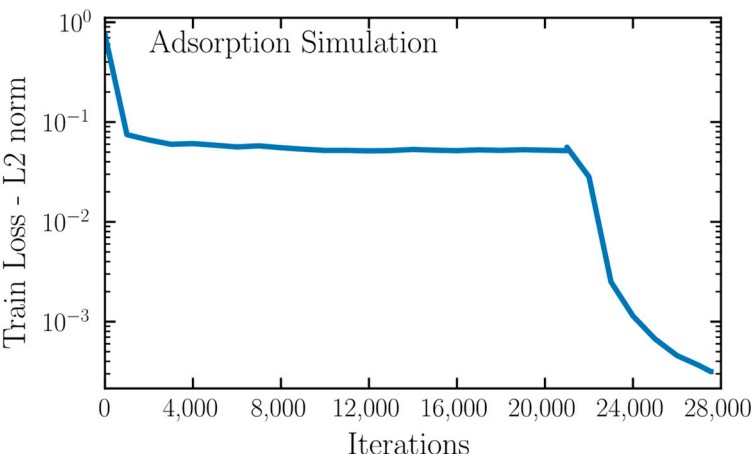

**Figure 3.** Train loss history for numerical adsorption experiment.

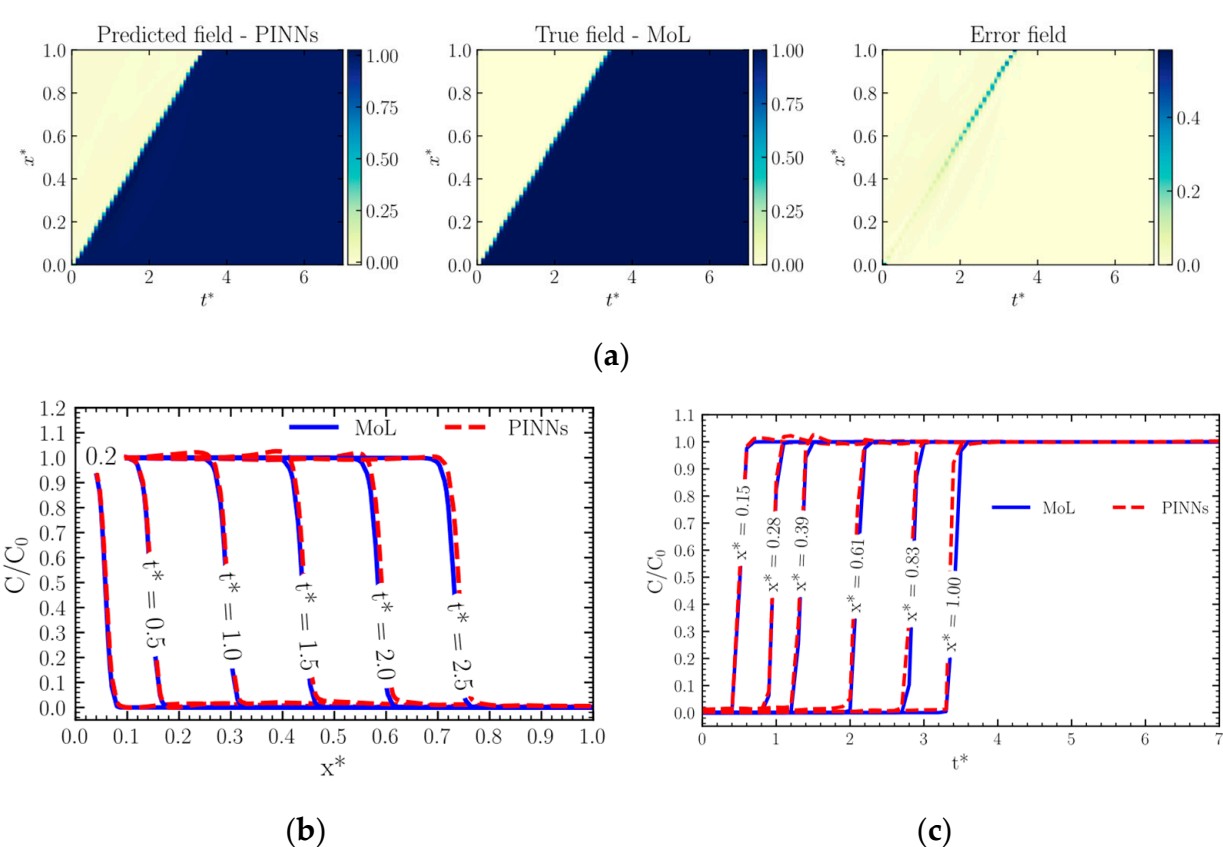

**Figure 4.** MoL and PINNs solution's comparison for adsorption experiment.

The numerical adsorption experiment is challenging to simulate for any numerical method. As known [18], neural networks are universal approximators for C1 continuous functions, and fields with steep gradients as those seen in adsorption are challenging. Even so, the PINN provides a satisfactory approximation to the PDE solution (mean absolute error calculated from a ground truth of less than 1%).

It is important to highlight that the PINNs are replacing the numerical method to solve the PDE system that composes the mechanistic model of the adsorption separation. Hence, it is preferred to compare its solution with a classical numerical method, the MoL, rather than experimental data. Furthermore, the model used as a reference was already

experimentally validated in the literature. The validation against experimental data can be achieved in future applications of this method in other systems, which would be an important development of this work.

Therefore, we certify that the PINNs can efficiently solve the numerical problem within a continuous domain. This also allows us to compare the computational effort of trained PINNs against the numerical solution by a classical numerical method. For instance, the MoL solved the model in 37 s. While the PINNs, once trained, provide a prediction in a fraction of a second. The comparison was made in an Intel Core i7-7500U CPU @ 2.70 GHz 2.90 GHz. This difference is significant in an optimization context, where hundreds of thousands of model evaluations are performed. In this way, the PINNs might allow the employment of mechanistic-informed models in an online context, such as real-time optimization, process control, and optimization. This can unlock several new possibilities in the field of process systems engineering. However, we are in an early stage of this technique, and further studies need to be carried out.

## 4. Conclusions

This work presents one of the first attempts to solve numerical problems arising from adsorption models with physics-informed neural networks. This is a very recent field of study and needs further studies to explore its full potential. This work provides the basis in this direction in chemical engineering, more specifically in the area of adsorption-based systems.

One key aspect to consider when comparing PINN with traditional numerical solvers is that the former does not require special treatment for issues arising in methods such as MoL, namely: a careful choice of spatial discretization scheme, index reduction methods, and appropriate time integrators for stiff systems. In a PINN solver, space-time points are sampled straightforwardly, and the problem is solved when the residuals reach an acceptable value (L2 norm in the order $1 \times 10^{-4}$ for conservation laws is enough for most practical applications). Moreover, the PINN can accept additional independent variables in the input layer apart from spatial and time, such as PDE, geometrical parameters, and initial and boundary conditions. Then, inference-time predictions can be delivered in a fraction of a second at any position and time within the training domain for several system configurations. Once the PINN is known to solve the PDE describing adsorption, it can be safely used for parameter estimation with simultaneous solution of the PDE and assimilation from sensor data and time-demanding tasks with traditional numerical solvers.

Therefore, the results of this work demonstrated that the PINNs are efficient in addressing the numerical solution of fixed-bed protein separation by adsorption. Both monoclonal antibody adsorption were calculated using the first-principles model solved using the physical-informed networks. The next step to explore this technique would be to evaluate more complex processes, such as continuous chromatography. From the current state of the literature and the present results, it is possible to envision the significant impact of the PINNs on the chemical systems phenomenological modeling.

**Author Contributions:** Conceptualization, I.B.R.N. and V.V.S.; formal analysis, V.V.S. and M.S.G.; writing—original draft preparation, V.V.S., I.B.R.N., M.S.G. and A.G.B.J.; review and editing, I.B.R.N., A.G.B.J., A.M.R., J.M.L., A.E.R. and F.W.T.; supervision, I.B.R.N., A.G.B.J., A.M.R., J.M.L., A.E.R. and F.W.T. All authors have read and agreed to the published version of the manuscript.

**Funding:** This work was financially supported by LA/P/0045/2020 (ALiCE), UIDB/50020/2020 and UIDP/50020/2020 (LSRE-LCM), funded by national funds through FCT/MCTES (PIDDAC). and FCT—Fundação para a Ciência e Tecnologia under CEEC Institucional program, and the National Council for Technological and Scientific Development (CNPq, Brazil) and the Coordination for the Improvement of Higher Education Personnel (CAPES, Brazil). The authors thank the financial support from the Brazilian National Agency of Petroleum, Natural Gas and Biofuels (ANP, Brazil) and PETROBRAS through the Clause of Investments in Research, Development, and Innovation.

**Institutional Review Board Statement:** Not applicable.

**Informed Consent Statement:** Not applicable.

**Data Availability Statement:** Not applicable.

**Conflicts of Interest:** The authors declare no conflict of interest.

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
