# Peer review of "A First Approach towards Adsorption-Oriented Physics-Informed Neural Networks: Monoclonal Antibody Adsorption Performance on an Ion-Exchange Column as a Case Study"

_2305-7084, doi:10.3390/chemengineering6020021_

Round 1

Reviewer 1 Report

The work entitled “A First Approach Towards an Adsorption-Oriented Physics-Informed Neural Networks: Monoclonal Antibody Adsorption Performance on an Ion-Exchange Column as a Case Study” performs the application of PINNs to a case study of the purification of a monoclonal antibody in a fixed-bed column. The results of the method using PINNs are compares with those of other numerical method.

The work is of interest, and the topic has large room to grow. However, I feel that there are some questions to be clarified before publication.

1 - The motivation of the work is not clear. It would be of interest to include in the introduction, why the authors are interested in this topic, i.e., the motivation?

2 – In section 2.2 two scenarios are mentioned, however, only one seems to be described. Perhaps it is not clear what the authors mean by scenario? Please, revise and clarify this section.

3 – Clearly, there is lack of experimental validation of this work. Please, discuss how this work can be translated to an effective practical application. Compare the results with experimental data, and/or explain the needs and future perspectives for experimental validation. This may be the missing section 4 that lacks in the structure of the manuscript.

4 - Line 171 - It should be "equations 16 to 19".

Author Response

Thank you for your comments. Please see attached file.

Reviewer 2 Report

This paper introduces a case to demonstrate an approach toward adsorption-oriented physics-informed-neural networks. It is very interesting and useful in the field of chemical engineering. Also, it is organized well and data strongly support their conclusions. I strongly recommend it to be accepted by this journal after considering the following points.

  1. Please explain more quantitative advantages of this approach compared to the present methods.
  2. In addition, it will be much better to add schematic illustrations of antibody adsorbed on  ion-exchange column.
  3. If possible, please give more relevant applications.
  4. Is there any limitations of this approach?

Author Response

(The authors gave the same response as above.)

Round 2

Reviewer 1 Report

Authors have answered convincingly to all comments.